# Using iPSC Models to Understand the Role of Estrogen in Neuron–Glia Interactions in Schizophrenia and Bipolar Disorder

**DOI:** 10.3390/cells10020209

**Published:** 2021-01-21

**Authors:** Denis Reis de Assis, Attila Szabo, Jordi Requena Osete, Francesca Puppo, Kevin S. O’Connell, Ibrahim A. Akkouh, Timothy Hughes, Evgeniia Frei, Ole A. Andreassen, Srdjan Djurovic

**Affiliations:** 1NORMENT, Institute of Clinical Medicine, University of Oslo & Division of Mental Health and Addiction, Oslo University Hospital, 0450 Oslo, Norway; attila.szabo@medisin.uio.no (A.S.); j.r.osete@medisin.uio.no (J.R.O.); fpuppo@health.ucsd.edu (F.P.); kevin.oconnell@medisin.uio.no (K.S.O.); ibrahim.akkouh@medisin.uio.no (I.A.A.); timothy.hughes@medisin.uio.no (T.H.); evgeniia.frei@gmail.com (E.F.); ole.andreassen@medisin.uio.no (O.A.A.); 2Department of Medical Genetics, Oslo University Hospital, 0450 Oslo, Norway; 3Department of Neurosciences, University of California San Diego, La Jolla, CA 92093, USA; 4Division of Mental Health and Addiction, Oslo University Hospital, 0372 Oslo, Norway; 5NORMENT, Department of Clinical Science, University of Bergen, 5020 Bergen, Norway

**Keywords:** schizophrenia, bipolar disorder, estrogen, neuron–glia interactions, iPS cells, brain organoids, drug target, pathophysiological mechanisms, in vitro model

## Abstract

Schizophrenia (SCZ) and bipolar disorder (BIP) are severe mental disorders with a considerable disease burden worldwide due to early age of onset, chronicity, and lack of efficient treatments or prevention strategies. Whilst our current knowledge is that SCZ and BIP are highly heritable and share common pathophysiological mechanisms associated with cellular signaling, neurotransmission, energy metabolism, and neuroinflammation, the development of novel therapies has been hampered by the unavailability of appropriate models to identify novel targetable pathomechanisms. Recent data suggest that neuron–glia interactions are disturbed in SCZ and BIP, and are modulated by estrogen (E2). However, most of the knowledge we have so far on the neuromodulatory effects of E2 came from studies on animal models and human cell lines, and may not accurately reflect many processes occurring exclusively in the human brain. Thus, here we highlight the advantages of using induced pluripotent stem cell (iPSC) models to revisit studies of mechanisms underlying beneficial effects of E2 in human brain cells. A better understanding of these mechanisms opens the opportunity to identify putative targets of novel therapeutic agents for SCZ and BIP. In this review, we first summarize the literature on the molecular mechanisms involved in SCZ and BIP pathology and the beneficial effects of E2 on neuron–glia interactions. Then, we briefly present the most recent developments in the iPSC field, emphasizing the potential of using patient-derived iPSCs as more relevant models to study the effects of E2 on neuron–glia interactions.

## 1. Introduction

Mental disorders are debilitating complex diseases, characterized by highly heterogeneous alterations in mood and behavioral traits, usually arising early in life in affected individuals [1,2]. The etiology of these disorders is complex and remains largely unexplained, hampering the development of preventive or curative treatments [3]. Schizophrenia (SCZ) predominantly affects perception, thoughts, and cognition, whereas bipolar disorder (BIP) mainly affects mood; however, there is a large degree of overlap in clinical characteristics and psychopharmacological treatment. Furthermore, subjects with SCZ and BIP have a significantly shorter life expectancy (e.g., due to comorbid somatic disease, accidents, or suicide). Thus, many patients remain chronically ill, making mental disorders one of the leading causes of the global burden of disease [4,5]. SCZ and BIP have a life time risk around 1% and 2%, respectively [6,7,8]. During the last decades, population and family-based genetic studies have demonstrated high heritability (60–80%) estimates for both SCZ and BIP, implicating genetic factors in disease etiology [9], with a large degree of overlapping genetic risk factors.

One common feature of both SCZ and BIP is the presence of sexual dimorphism with regards to prevalence, age of onset, symptom severity, and response to drug treatments [10,11,12], as well as life expectancy [8,11]. Men have nearly 40% higher risk of developing SCZ than women [11,13]. There is also evidence that the prevalence of BIP type I is higher in men, whereas BIP type II is more prevalent in women [14]. Male patients with BIP usually present with earlier onset than female patients and are more prone to develop mania, and comorbid substance abuse as compared to female patients with BIP, who have a higher tendency to present with comorbid disorders of panic, eating, post-traumatic stress, and borderline personality [15]. A recent and large population-based study from Sweden suggests that the prediction accuracy of polygenic risk scores (PRS) is significantly lower in female participants with SCZ, compared to males [16].

In women, there is an increased risk of cognitive decline and dementia after menopause, a period of a drastic decrease of estradiol levels (17β-estradiol, E2) [12,17], suggesting that E2 may act as a physiological neuroprotector factor against age-related neurodegenerative disorders [12,18]. Conversely, the Y chromosome gene *SRY* may act as a potential risk factor for neurological disturbances in men [12], and suppressing its expression has conferred neuroprotection in animal and cellular models of neurodegeneration [19]. Regarding severe mental disorders, women with SCZ tend to respond better to antipsychotic therapy as compared to men (recently reviewed in Reference [20]). Moreover, in women affected with SCZ, decreases in the severity of psychotic symptoms correspond to periods of increased E2 levels during the hormonal cycle, and vice versa, suggesting a neuromodulatory role of E2 [12]. These observations resulted in the estrogen hypothesis of SCZ, which states that E2 may provide protection from the development of SCZ [11,21].

Combining E2 or selective estrogen receptor modulators (SERMs) with antipsychotics or mood stabilizers have been successful therapeutic strategies against SCZ and BIP symptoms [11,22,23]. This underscores the importance of a more detailed understanding of the mechanisms underlying the neuromodulatory effect of E2 in mental illnesses, which remains largely unknown, and holds the potential for improved treatments for both male and female patients with SCZ and BIP [24].

In this regard, there is increasing evidence that the mechanisms by which E2 modulates neuronal activity are highly dependent on glial cells, since both neurons and glia express different types of E2 receptors (ERs), where E2 binds to its cognate receptors directly or indirectly and influences the function and fate of neurons through neuron–glia interactions [25,26,27]. However, studying neuron–glia crosstalk in mental disorders is a challenging task given the limitations of accessing human brain tissue and the difficulties of translating the complexity of the human brain to animal and cell line models [28]. One promising approach is the use of induced pluripotent stem cells (iPSCs), which can be obtained by reprogramming somatic cells from patients [28]. IPSCs derived from SCZ and BIP patients have been routinely generated and differentiated into disease relevant cell types such as neural progenitor cells (NPCs) [29], astrocytes [30] and neurons [31]. IPSCs can also be grown into three-dimensional (3D) structures known as organoids, or “mini-brains”, which resemble specific brain regions [32,33]. The advantage of this model system is that it enables us to study both neuronal networks similar to those in the human brain, as well as interactions between glia and neurons.

In this review, we first describe the specific ERs present in the human brain, and then present an update of recent genetic findings in SCZ and BIP, with a particular focus on results from genome-wide associations studies (GWASs). Next, we give a brief overview of key molecular aspects related to the pathophysiology of SCZ and BIP, including brain energy metabolism, neuroinflammation, and neurotransmission. Then, we discuss the involvement of E2 in these processes. Additionally, we briefly review the current state of the art in iPSC-based technologies and indicate how these tools can be utilized to study neuron–glia interactions and to elucidate mechanisms underlying the beneficial effects of E2 in SCZ and BIP.

## 2. Estrogen and Estrogen Receptors in the Brain

The pool of E2 in the brain is maintained by E2 synthesized in the gonads, breasts and also by local production, since the enzyme aromatase, which converts androgens into E2, is present in the brain [34]. E2 acts through genomic and non-genomic mechanisms. ERs can act as ligand-activated transcription factors by directly binding to estrogen response elements (EREs) sequences through classical signaling mechanisms [35], enabling E2 to produce a wide range of transcriptional changes in different tissues and cell types. Genomic signaling is mediated by nuclear receptors in the cellular cytoplasm. In this case, activated ERs take a longer time to translocate to the nucleus and cause transcriptional activation, leading to a prolonged action. Non-genomic mechanisms result in a faster effect, since E2 is mediated by ERs at the plasma membrane and acts through second messenger cascades [36]. The most studied ERs are the ERα, encoded by the estrogen receptor 1 (*ESR1*) gene, and the ERβ, encoded by the estrogen receptor 2 (*ESR2*) gene, both expressed in the human brain. E2 has greater affinity for ERα than for ERβ [37]. Both ERs are present in the human prefrontal and temporal cortex, where ERα is more abundant in the nucleus and ERβ in the cytoplasm, but it seems that the ERβ is not present in the layer I of the temporal cortex [38,39]. However, in the human brain, ERβ is the predominant ER type in the hippocampus [39,40], and thalamus [41], whereas ERα predominates in the amygdala and hypothalamus [42]. Nevertheless, the fact that ERβ is the predominant estrogen receptor type in the adult brain of rats and humans [39], whilst in the adult mouse brain ERα predominates in the hippocampus and is almost absent in the cerebral cortex and cerebellum [43], indicates that there are species differences in the brain expression of ERs, and caution should be taken in extrapolating results from animal models to humans.

In addition to the steroid hormone receptors ERα and ERβ, a G-protein-coupled receptor superfamily member, which is synthesized in the endoplasmic reticulum and denominated either GP30 or G-protein estrogen receptor (GPER), has been identified. Upon interaction with E2, GPERs undergo conformational changes inducing the conversion of guanosine triphosphate (GTP) to guanosine diphosphate (GDP) within its Gα subunit, which results in intracellular production of cyclic adenosine monophosphate (cAMP) and rapid estrogen-mediated actions. GPER is widely expressed in the central nervous system (CNS), especially in the hypothalamic pituitary axis, hippocampus, and brain stem autonomic nuclei [44]. Finally, researchers have identified a putative ER, denominated ER-X, which is characterized by its equal binding affinity for both 17α- and 17β-estradiol, contrary to ERα and ERβ, which have 100 times greater binding affinity for 17β-estradiol than for 17α-estradiol. ER-X is found within the plasma membrane of cells from the neocortex. ER-X is upregulated in the neocortex of a mouse model of brain ischemic injury, and since ER-X expression is developmentally regulated in the brain, with peak expression at postnatal days 7–10 and reduced levels with age in adult mice, it has been suggested to play a role in early stages of brain development [45].

In patients with SCZ, susceptibility SNPs in one intron of the *ESR1* gene have been reported, and are associated with decreased mRNA expression of ERα in the frontal cortex [46,47]. Decreased ERα expression has also been reported not only in the hippocampus of individuals with SCZ [48], but also in the hippocampus of ovariectomized rats, where this effect is prevented by E2 treatment [49], suggesting that, at least in the case of female rats, decreased E2 levels could downregulate hippocampal ERα expression. In intact and ovariectomized rats, almost all of the ERβ-positive cells in the brain cortex and hippocampus co-localize with the protein parvalbumin (PV), characteristic of gamma-aminobutyric acid (GABA)ergic inhibitory neurons, suggesting a physiological role of E2 in the modulation of memory and cognition by controlling the inhibitory tone in the mentioned brain regions [49]. This co-localization of ERβ with hippocampal GABAergic neurons may explain, at least in part, the fact that hippocampal GABAergic neurons are more affected in men than in women with SCZ [50] and the fact that, in general, women with SCZ experience less cognitive problems than men [51], since these cells produce gamma-band oscillations that are essential for cognitive functions [52]. GPER has been implicated in hippocampal neuroprotection in a rat model of global ischemia. When the animals were treated with E2 or with the non-classical ER agonists diphenylacrylamide or G1, the CA1 region of the hippocampus was protected, and neuroprotection was abrogated by an antisense knockdown of GPER [44]. The activation of GPER has also been associated with increased levels of the antiapoptotic proteins Akt and Erk 1/2, neuroprotective effects relevant for neurodegenerative diseases, mechanisms of energy control in hypothalamic neurons, modulation of cholinergic function and cognitive performance in the basal forebrain, and seems to play a role in anxiety and stress responses [44].

## 3. Genetics of Schizophrenia and Bipolar Disorder

Genetic factors contribute substantially to the pathomechanisms underlying both SCZ and BIP. Twin studies have consistently found high heritability estimates for both disorders: a meta-analysis of 12 published twin studies of SCZ found that the heritability was 81% (95% CI: 73–90%) [53]. A more recent twin study, which is the largest to date, comprising 31,524 twin pairs, found a similarly high heritability (79%) [54]. Heritability estimates for BIP are nearly as high as SCZ and range from 59 to 87% [55]. A recent population-based twin study found the heritability of BIP to be 60.4% [56]. A Swedish family based study found the heritability for SCZ and BIP was 64% and 59%, respectively [9].

The Psychiatric Genomics Consortium (PGC), a large international collaboration [57], has provided large-scale GWAS data to decipher the genetic architecture underlying SCZ [58] and BIP [59]. Several works from this consortium have identified strong associations between multiple common genetic variants of small effects and SCZ and BIP, and have suggested that SCZ and BIP are neurodevelopmental disorders that not only share clinical symptoms, but also display a substantial genetic overlap [9]. For a complete review of the polygenic architecture of SCZ, see Reference [60].

A landmark GWAS by the PGC, using a sample of 36,989 cases and 113,075 controls, identified 108 loci associated with an increased risk for SCZ, 75% of which include protein-coding genes, many being expressed in the brain [61]. Recently, the PGC Working Group on SCZ performed analysis on 69,369 SCZ patients and 236,642 controls and identified common variant associations at 270 distinct loci [62]. Fine-mapped candidates were enriched for genes associated with rare disruptive coding variants in SCZ patients, including the glutamate receptor subunit GRIN2A and the transcription factor SP4, as well as genes implicated in autism and developmental disorder. Interestingly, there was a convergence of common and rare variant associations in SCZ and neurodevelopmental disorders [62].

Other GWASs have identified a large number of susceptibility loci associated with mental disorders, including 30 for BIP [59] and 44 for major depressive disorder [63]. Genes located within the associated genomic regions are highly enriched for biological pathways involved in neurodevelopment, cellular signaling and immunity, such as ion channel signaling, synaptic function and neurotransmitter pathways [64,65,66]. Recently, a GWAS including 41,917 BIP cases and 371,549 controls identified 64 associated genomic loci. These BIP risk alleles were particularly enriched for genes expressed in neurons of the prefrontal cortex and hippocampus involved in synaptic and calcium signaling pathways [67].

In the past five years, large transcriptomic datasets derived from multiple tissues, cell lines, and organisms have been conducted and dysregulation of hundreds of genes in different brain regions has been reported in SCZ [60,68,69,70]. Moreover, it has been shown that patterns of transcriptional dysregulation are shared across major psychiatric disorders, confirming a genetic overlap between psychiatric disorders also at the transcriptome level [71].

Additionally, a recent proteomic analysis review of postmortem brain tissue showed alterations in 92 energy metabolism proteins in SCZ and in 95 energy metabolism proteins in BIP. Of these proteins, 32 were shared between SCZ and BIP, and most of them related to mitochondrial electron transport, response to reactive oxygen species and glycolysis [72]. Interestingly, immunological status, energy metabolism, and neurotransmission exert strong influence on each other and are highly dependent on neuron–glia interactions [73].

GWASs from European descents [74] and from the Korean population [47] have identified associations of SNPs related to E2 biosynthesis and ERs with SCZ. More recent GWASs have identified that a genetic risk (E2 polygenic risk score) for higher plasma E2 is negatively associated with hippocampal volume, but not with an increased risk of major depressive disorder or postpartum depression [75]. However, the largest GWAS of anxiety traits to date was able to identify genome-wide significant associations near genes involved with the ERα (ESR1) [76]. Moreover, the SNP rs2144025 has been associated with increased ESR1 mRNA levels in prefrontal cortex from subjects with BIP and SCZ and appears to modulate traits in behavioral disorders [77].

## 4. Brain Energy Metabolism in Schizophrenia and Bipolar Disorder

### 4.1. Brain Energy Metabolism Overview

Despite the fact that the human brain accounts for only 2% of the body weight, it expends around 20% of the total energy produced in the whole body [78]. This is due to the high energy needs of neuronal cells for neurotransmission, including the formation of action potentials, vesicular neurotransmitter transport and in the restoration of ion gradients after neuronal depolarization by the Na^+^/K^+^-ATPase pump, in particular at the presynaptic terminals [78,79]. The brain tissue relies highly on the adenosine triphosphate (ATP) produced via mitochondrial oxidative phosphorylation (OXPHOS) [78]. The electron transport chain (ETC) is constituted of four enzyme complexes embedded in the mitochondrial inner membrane (complexes I-IV), whose function is to transfer electrons from energy substrates, such as glucose, pyruvate, and lactate, to the final electron acceptor, oxygen. There is a concomitant transfer of protons (H^+^) into the mitochondrial matrix by the complexes I, III and IV, generating a proton gradient that provides the energy necessary for the phosphorylation of adenosine diphosphate (ADP) into ATP when the complex V (ATP synthase) transports the protons from the mitochondrial matrix back to the mitochondrial intermembrane space. During the OXPHOS process, oxygen reactive species (ROS) are formed as a byproduct of the mitochondrial respiration by the complexes I, II and III of the ETC [80]. Besides acting as a cellular power plant to produce ATP, mitochondria also act as important calcium (Ca^2+^) buffer for cells, regulating cellular oxidative stress, and controlling cell fate and the synthesis of steroid hormones and growth factors in the CNS [78].

The main source of energy for the brain under normal physiological conditions is glucose, whilst lactate, triglycerides and ketone bodies can also be used under special conditions [81]. Astrocytes have a predominantly glycolytic metabolism, producing ATP from glucose under aerobic conditions and releasing lactate to the extracellular space. Astrocytes are also able to use fatty acids as an energy source and to store glycogen [81,82]. Neuronal activity demands intense mitochondrial activity, along with accumulation of neurotransmitters in the terminal synaptic cleft, especially glutamate, making neurons highly susceptible to ROS damage and glutamate excitotoxicity [52]. This is the case for PV GABAergic interneurons, fast spiking inhibitory neurons that produce large amounts of ROS, which present a high expression of Ca^2+^ permeable AMPA receptors, and are implicated in SCZ [52]. Since glucose is important to maintain the levels of glutathione, an important ROS scavenger and precursor of glutamate, it has been postulated that, contrary to astrocytes, neurons spare glucose as a neuroprotective strategy and preferably use lactate as the main source of energy via OXPHOS [78]. Accordingly, since the 1990s the astrocyte–neuron lactate shuttle (ANLS) hypothesis has postulated that in the presence of high glutamate concentrations, astrocytes are stimulated to produce lactate from glucose and secrete it as a source of energy for neurons [78,83,84]. Despite being widely accepted, there is some debate over the ANLS hypothesis since neurons are able to metabolize other energy substrates than lactate [85,86].

### 4.2. Brain Energy Metabolism in SCZ and BIP

Proteomics studies in postmortem brain tissue have shown that SCZ and BIP share common alterations in more than 30 proteins, most of them related to mitochondrial ETC (complexes I and V), antioxidant defense (peroxiredoxins, glutathione S-transferase and superoxide dismutase), glycolysis (pyruvate kinase and fosfofrutokinase), and ATP transfer from the mitochondria to other cellular compartments (creatine kinase) [72]. This is in agreement with several cellular and rodent models of SCZ and BIP finding associations with mitochondrial malfunctioning and oxidative stress [87,88,89,90,91,92,93].

In addition, decreased levels of the subunit NDUFS7 of complex I have been found in postmortem brain tissue from patients with BIP, suggesting an inhibition of complex I activity, and increased protein carbonylation, a marker of oxidative stress [94]. A more recent study used hippocampal-dentate-gyrus-like neurons derived from iPSCs of BIP patients to model BIP. These cells had increased mitochondrial gene expression, increased mitochondrial membrane potential, decreased mitochondrial size, and neuronal hyperexcitability [31]. A study of postmortem brain tissue in patients with SCZ revealed a 43% reduction in COX (ETC complex IV) activity in the cortex gyrus frontalis and a 63% reduction in the caudate nucleus [95]. Another study on postmortem brain from patients with SCZ found decreased complex IV activity in the caudate nucleus but increased activity in the putamen and nucleus accumbens of the brain, especially in patients that have suffered from paranoia. However, the authors could not rule out the influence of neuroleptics treatment on these effects [96]. Furthermore, a study investigating mitochondria isolated from blood platelets from patients with SCZ, BIP and major depressive disorder showed a 240% increase in complex I, but not in complex II activity, in SCZ patients compared to controls. Interestingly, both medicated and non-medicated SCZ patients showed similar results [97]. Finally, a more recent study from Ni et al. (2019) tested mitochondrial function and gene expression in iPSC-derived cortical interneurons (cINs) and glutamatergic neurons from 15 healthy controls and 15 patients with SCZ. Only SCZ cINs, but not SCZ glutamatergic neurons, showed both decreased expression of mitochondrial genes and decreased mitochondrial respiration. The genes *ND2* and *ND4L*, implicated in the OXPHOS complex I, were downregulated, and maximal respiration and reserve capacity were decreased. Mitochondrial dysfunction was followed by an increase in ROS and reduced arborization, only in the cINs, but not in the glutamatergic neurons, suggesting that mitochondrial dysfunction is intrinsic to cortical interneurons in SCZ. Both the mitochondrial hypofunction and decreased arborization were reversed by treating cINs with acetyl-L-carnitine, a well-known mitochondrial modulator, indicating that targeting mitochondrial dysfunction in SCZ cINs may be utilized as a therapeutic intervention for certain clinical symptoms of SCZ, such as alterations in gamma oscillations and cognition [98].

Interestingly, patients diagnosed with mitochondrial disorders often present with psychiatric symptoms, and drugs used in the treatment of SCZ and BIP modulate mitochondrial metabolism [99,100]. Altogether, the aforementioned evidence strongly suggests that mitochondrial metabolism and oxidative stress play an important role in the pathophysiology of SCZ and BIP. Besides, there is evidence that mitophagy, the selective elimination of malfunctioning mitochondria which is essential for maintaining cellular viability, may be dysregulated in several psychiatric disorders [101]. In postmortembrains of SCZ patients several key genes involved in neuronal autophagy were downregulated and correlated with positive symptoms in SCZ [102]. In a microtubule-associated protein 6 (MAP6)-deficient mouse model of SCZ, the behavioral phenotype was ameliorated by the administration of davunetide, a peptide which enhances autophagy. The symptoms were completely rescued by a combination of davunetide and clozapine, indicating that autophagy plays a role in SCZ [103]. In contrast, the impaired myelination followed by decreased oligodendrocyte densities and morphological alterations in these cells have been attributed to increased oligodendrocyte autophagy in SCZ. Whilst a considerable amount of oligodendrocyte-associated genes are downregulated in SCZ, *DISC1* and *PHB2*, which encode for mitophagy receptors at the inner mitochondrial membrane, are upregulated. Moreover, the abrupt reduction in the number and densities of oligodendroglial mitochondria and the normal appearance of the remaining mitochondria point to an enhanced mitophagy in oligodendrocytes in SCZ [104]. In patients with BIP, proteins related to mitophagy in blood cells were downregulated, accompanied by decreased mitochondrial ATP production and increased ROS levels [105]. Dysregulations of the mitophagic pathway leads to the accumulation of damaged mitochondria, resulting in increased oxidative stress, decreased mitochondrial Ca^2+^ buffering capacity and loss of ATP, which are particularly harmful in post-mitotic cells such as neurons [105]. One could speculate that in BIP patients, the number of damaged mitochondria may exceed the capacity of mitophagy, and apoptosis may become the dominant pathway to minimize tissue damage. In addition, a cell non-autonomous cytoprotective mechanism, transcellular mitophagy, is able to mediate the removal of defective mitochondria, in which a pool of axonal mitochondria are transported to and degraded by adjacent astrocytes. This process could be dysregulated in SCZ and BIP [101]. Thus, it is conceivable that alterations in astrocytic transcellular mitophagy could take place in SCZ and BIP. Interestingly, several antipsychotic drugs, such as fluspirilene, trifluoperazine, and pimozide, and the mood stabilizer lithium are inducers of autophagy [102,106], while haloperidol and clozapine block autophagy by inhibiting the formation of autophagolysosomes [106].

Another important feature that is common to both SCZ and BIP is the presence of an imbalance between proteins involved with the increase of intracellular Ca^2+^ and those responsible for restoring Ca^2+^ levels [107]. Indeed, some antipsychotic drugs target either the calmoduline pathway [108] or inositol-3-phosphate (IP3)-induced Ca^2+^ release [109]. In platelets of patients with BIP, thapsigargin, a drug that induces Ca^2+^ release from the endoplasmic reticulum, elicits a greater Ca^2+^ release in patient platelets than in those from controls [110], whereas platelets from patients with SCZ show increased cytosolic Ca^2+^ and calcium-related alterations in the distribution of membrane phospholipids [111]. This is in line with the fact that among the most consistent GWAS findings are associations between SNPs in the α1 subunit (CACNA1C) of the voltage-gated L-type Ca^2+^ channel (LTCC) Cav1.2 and SCZ and BIP [112], and that several proteins related to Ca^2+^ metabolism are found altered in individuals with SCZ [113]. These Ca^2+^ channel alterations may impact mitochondrial metabolism, since modulating LTTC or the *CACNA1C* gene prevents mitochondrial ROS overproduction, disruption of mitochondrial membrane potential, loss of ATP, decrease in mitochondrial respiration, and oxidative cell death in neuronal cells [114]. Besides producing ATP and ROS, and mediating the effects of glutamate and of brain-derived neurotrophic factor (BDNF) on neural plasticity, mitochondria have a key role in buffering calcium and regulating apoptosis [99]. Thus, even in the absence of Ca^2+^ signaling-related SNPs, Ca^2+^ imbalance in SCZ and BIP could also result from a downstream effect of dysfunctional mitochondria, since disturbed mitochondria might fail in the task of buffering cytosolic Ca^2+^ [93].

### 4.3. Influence of Estrogen on the Mitochondrial Metabolism

E2 modulates mitochondria in several ways, including the expression of ETC complexes and ATP synthesis, ROS production and antioxidant defenses, apoptotic signaling pathways, and Ca^2+^ homeostasis in both physiological and pathological conditions. The classical model of ovariectomized mice has been used to test the effects of physiological concentrations of E2 or specific E2 receptors on brain mitochondrial activity. Findings suggest that the stimulation of both ERα and ERβ leads to an increase in complex IV (COX) activity and in the expression levels of nuclear genes encoding the COX-IV subunit. ERβ activation also promotes the expression of the mitochondrial genes encoding the COX-I subunit. Both ERs increase the expression of the antioxidant enzymes manganese superoxide dismutase (MnSOD), peroxiredoxin V (PrdxV) and phospholipid hydroperoxide glutathione peroxidase (PhGPx4), resulting in reduced lipoperoxidation, and both ERs also stimulate the maximal mitochondrial capacity in neurons and glia [115]. The activation of specific nuclear ERs by E2 alters mitochondrial metabolism by modulating the expression of respiratory chain complexes and other mitochondria associated proteins. For example, the ER isoform ERβ1, but not the isoforms ERβ2 and ERβ3, is associated with mitochondrial protection against oxidative stress and mitochondrial membrane permeability in human lens epithelial cells [116]. E2 treatment has been associated with increased expression of the cytochrome c oxidase subunits II [117], III [118], and VII [119], with increased activity of the enzyme cytochrome c oxidase (ETC complex IV) [118], and with the ATP synthase (ETC complex V) [120]. These effects possibly result from a direct regulation of mitochondrial transcription, provoking an increase of the mitochondrial content in the cells, since E2 provokes an increase in the ratio of mitochondrial DNA (mtDNA) over nuclear DNA (nuDNA) [120]. Another important target of E2 is the nuclear respiratory factor 1 (NRF-1), a mediator of mitochondrial activity and stimulator of mitochondrial biogenesis via ERα [121]. E2 is able to modify the morphology and structural integrity of mitochondria [122], and to increase the transcription of proteins for mitochondrial fusion and fission in both male and female astrocytes [123]. Since astrocytes are essential for the nutrition of neurons, it is believed that sex differences of E2 mitochondrial modulation in astrocytes may explain, at least in part, gender differences in several brain cellular pathologies [122].

E2 promotes cell survival signaling by increasing Ca^2+^ influx through the mitochondrial calcium uniporter (MCU) by an ERα-dependent mechanism [124]. E2 also enhances the expression of anti-apoptotic proteins such as B-cell lymphoma 2 (Bcl-2) [125] and antioxidant proteins, such as thioredoxin, manganese superoxide dismutase [126], and glutathione [127]. Bcl-2 family proteins regulate the import of cytosolic Ca^2+^ into the mitochondria, contributing to cellular Ca^2+^ buffering, and preventing the activation of the permeability transition pore by cytosolic Ca^2+^-overload. E2 also prevents caspase-3 activation and mitochondrial cytochrome c release in models of glutamatergic excitotoxicity [128]. The SERMs tamoxifen and raloxifene reduce oxidative stress by increasing the expression of antioxidant defense enzymes [22]. The antioxidant effect of raloxifene includes regulation of Bcl-2, catalase, superoxide dismutase, and glutathione peroxidase gene expression and the level of reduced glutathione in the brain [22]. Additionally, raloxifene increases mRNA expression of apurinic/apyrimidinic endonuclease/redox factor-1 suggesting that it may protect against ROS-induced DNA damage [22]. Experiments in which primary cultured astrocytes are first deprived of oxygen and glucose and then treated with selective ER agonists suggest that the protective effects of estrogen on mitochondrial function, cellular death, and ATP production are mediated by ERα activation [129]. More recently, studies using neuronal and organotypic slice cultures from ERβ-knockout mice attributed E2 neuroprotection against glutamatergic excitotoxicity and oxygen glucose deprivation to a mechanism mediated by ERβ, which is able to prevent the mitochondrial permeability transition (MPT) and Ca^2+^ toxicity [130]. Taken together, the aforementioned works suggest a number of ways by which E2 may improve mitochondrial activity in several regions of the CNS. However, studies of modulatory mechanisms of E2 on mitochondria of human brain cells are still lacking, but hold the potential to confirm many of the routes identified in cell lines and rodent models. Such studies may also identify new mechanisms underlying the beneficial effects of E2 in several neurological conditions where gender differences exist, such as mental disorders.

## 5. Neuroinflammation in Schizophrenia and Bipolar Disorder

### 5.1. Neuroinflammation Overview

The complex interactions of the brain and immune system have been shown to be involved in the development and organization of the CNS tissue microenvironments, as well as in neuronal survival and cortical functions, such as higher-order cognitive processes in health and disease [131,132,133]. Dysregulation of the brain-immune axis can bring about a variety of pathologies, and manifestations of these abnormalities are usually dependent on the cellular and intracellular systems involved. Neuroinflammation is one of the most prominent and well-studied physiological consequences of a dysfunctional CNS-immune relationship, and the neuroinflammatory aspects of neuropsychiatric disorders are an emerging topic in biomedical research [133,134,135,136]. Interestingly, sex may underlie the neurodegenerative and inflammatory hallmarks of these disorders and recent reports suggest the functional engagement of glial cells, especially microglia, in the process [137].

Moreover, new evidence shows that variation of the complement component 4 (C4) genes *C4A* and *C4B*, which have been linked to increased risk for SCZ, contribute to sex-biased vulnerability: C4 alleles act more strongly in men than in women, implicating the complement system as a source of sexual dimorphism in vulnerability to SCZ [138]. Unlike SCZ, BIP appears to be less strongly characterized by gender-based pathophysiological dissimilarities with regards to neuroinflammation. However, aberrant and chronic inflammatory dysregulation of glial functions have also been documented in BIP [139].

This section aims to briefly summarize the causative role of different glial cells in neuroinflammation and the inflammatory gliopathy aspects of SCZ and BIP, as well as to discuss the possible influence of sex hormones and estrogen-related neuroendocrine and immune effects on glial cell functions and neuron–glia crosstalk.

### 5.2. Immune Responses and Inflammatory Glial Functions in SCZ and BIP

The immune system is an evolutionally advanced defense mechanism whose main role is to protect and conserve the integrity of the “biological self” from invading microbes and different forms of endogenous malignancies. The two major branches of the immune response are innate and adaptive immunity, which represent an ancient, germline-encoded, non-specific, rapid, and an adaptive, slower and highly antigen-specific response mechanism, respectively [140]. The secretion of inflammatory cytokines (e.g., interleukins, such as IL-1β, IL-6, and tumor necrosis factor-alpha, TNF-α) in response to pathogenic stimuli is an integrated part of the innate immune response, while the production of antibodies and the generation of epitope-specific immune memory are largely based on the cells of adaptive immunity [141]. Associations between SCZ, inflammation, and immune system activity were first reported by multiple studies, and later supported by epidemiological data suggesting an intricate causal connection between maternal infections, systemic inflammation, and altered cognitive functioning [142,143,144,145,146]. Early childhood autoimmune conditions have been associated with increased likelihood of psychotic episodes and SCZ in adults [147]. In addition, overall lifelong SCZ risk shows a strong positive correlation with the number of severe infections in people with autoimmune background [147]. The observed comorbidity tendencies between SCZ, infections and autoimmune disorders suggest a common underlying inflammatory component and have been hypothesized to involve the immunocompetent cells of the human brain: microglia and astrocytes [30,133,148].

Although microglia and macrophages are functionally similar and part of the mononuclear phagocyte system, macrophages derive from the myeloid–macrophage lineage whereas microglia derives from the “primitive” or “primordial” c-Kit^lo^CD41^lo^ erythroid progenitors in the yolk sac [149,150,151]. The ramified microglia niche of the adult brain expresses typical markers, such as CD11b, the fractalkine receptor CX3CR1, and the ionized calcium-binding adapter molecule 1 (Iba1) [152]. They are strongly involved in CNS immuno-surveillance, as well as in the neuroinflammatory processes of various neuropsychiatric disorders [151,153]. Neuroinflammation is characterized by region-specific activated microglia that upregulate the expression of the translocator protein (TSPO). Several neuroimaging studies using TSPO ligand reported neuroinflammation in both recent-onset SCZ [154] and in acute SCZ-related psychosis [155]. These studies also suggest that microglia-related neuroinflammation may be responsible for cortical and hippocampal atrophy, and cognitive decline in SCZ. An alternative explanation also involves activated astrocytes that likewise express TSPO, and that have also been documented in SCZ [133]. An interesting cell-type-intrinsic property of microglia is that an already primed (i.e., previously activated) glial cell can respond to new stimuli in a much stronger and rapid manner [156]. They may also form an immune memory of neuropathologies that can consecutively raise their responsiveness to new inflammatory insults [157]. This may partly support the developmental hypothesis of SCZ, as early (perinatal and/or childhood) infections and systemic inflammation might pose a stimulatory effect on microglia increasing glial activation and psychosis risk following later infections in adulthood.

The involvement of microglia-driven inflammation in the neuropathology of BIP was first characterized by human positron emission tomography (PET), where the authors described significantly increased in vivo microglia activity and consequent neuroinflammation in the hippocampus of BIP patients relative to healthy individuals [158]. In a follow-up study, the same group found a causal connection between increased microglial activity and neuronal damage in the affected brain regions in BIP [159]. Besides microglia, the oligodendrocyte compartment of the brain has also displayed abnormalities, such as decreased myelination of axons and significantly reduced cell numbers in postmortembrain samples from individuals with BIP [160,161]. In line with this, studies have reported strongly downregulated mRNA levels of oligodendrocyte-specific markers accompanied by mitochondrial and inflammatory dysregulation in BIP, raising the possibility of impaired axonal myelination, maintenance and repair in the brain of patients (reviewed in Reference [162]). It is important to note that chronic cellular stress in oligodendrocytes has been linked to the release of microglia-activating cytokines and chemokines that may lead to uncontrolled neuroinflammation [163], a process that has been suggested to play a role in neurobehavioral changes in BIP [139]. In addition, recent hypotheses on the alteration of peripheral immune functions and immune–glia crosstalk involve early T lymphocyte defects in BIP, suggesting that the disorder shares a similar neuro-immune signature with major depressive disorder (MDD) [164]. Mounting evidence shows that BIP pathophysiology may include early abnormalities in monocyte and T lymphocyte networks, similarly to those seen in MDD, which, however, can become in part restored with older age [139,165]. Furthermore, like SCZ and MDD, BIP shows significant comorbidity with systemic autoimmune pathologies, in which patients with existing autoimmune disorder(s) have an increased risk of developing BIP later in life, supporting the immune-brain inflammatory link in the etiology of the disease [166,167].

Astrocytes are characteristic neurosupportive glial cells of the CNS that also mediate inflammatory and immune functions [168,169]. Recent reports in astrocyte in vitro and in vivo experimental models, as well as in human postmortem brain samples, demonstrate considerable alterations in neuropsychiatric disorders, such as decreased astrocyte-specific functions and gene-signatures in major depression and anxiety, and increased neuroinflammation in SCZ [170]. Moreover, aberrant inflammatory astroglia functions have also been linked to prefrontal cortex atrophy and cognitive impairment in SCZ [171]. Resident glia-related neuroinflammation can be a consequence of dysregulated inflammasome activation. Inflammasomes are important elements of the innate immune system representing the early phase of response against pathogens by the release of the pro-inflammatory cytokines IL-1β and IL-18 via caspase 1 activation [172,173,174]. Recent studies demonstrated that the elevation of serum and postmortem brain IL-1β levels is associated with symptom severity and disease evolution in SCZ [175,176]. These inflammatory cytokines are essential in the activation of multiple downstream immune-effector pathways, and have also been associated with SCZ pathophysiology via their hypothetically aberrant secretion by glial cells [133]. Interestingly, multiple aspects of this innate immune activation and glia-mediated neuroinflammation exhibit gender differences at both the cellular and systemic levels [177,178].

### 5.3. The Effect of Sex Hormones on the Inflammatory Responses of Glial Cells: What Do We Know so Far?

Due to the observed sex differences in the prevalence of inflammatory pathologies, and the considerable variation in sensitivity in inflammation-related degenerative disorders across the lifespan, recent research efforts focus on sex differences in immune responses, neuroinflammation, and neuronal cell death pathway signaling [178,179,180,181,182]. Sex hormones influence neuronal death pathways, the neurotrophic activity of glial cells, and can also modulate the immune system at multiple levels. Various lymphoid (e.g., T cells, B cells, and natural killer cells) and myeloid cells (e.g., monocytes and macrophages), as well as microglia and astrocytes express ERs [183,184]. It is important to note that the effect of sex hormones on immune modulation can be dose-dependent. E2 has recently been shown to increase the immunoreactivity of microglia to inflammatory cues in the peripubertal sensitive period of female mice [185]. The authors showed that microglia in the ventromedial hypothalamus displayed greatly increased inflammatory cytokine responses, and acquired immunoreactive morphological features in the presence of E2 following lipopolysaccharide (LPS) challenge [185]. For instance, low doses of E2 favor Th1 (inflammatory) responses while high doses induce Th2 (anti-inflammatory) skewing in CD4^+^ helper T cell differentiation [149,181]. Unlike E2, progesterone uniformly downregulates the production of TNF-α and other inflammatory cytokines by exerting a direct inhibitory effect on the key transcription factor nuclear factor kappa (BNf-κB) [186,187]. Progesterone also modulates neurotransmission and oligodendrocyte and microglia activation, and thus promotes neurorestorative functions (e.g., myelin repair) [183,188]. Androgens, on the other hand, can both decrease and exacerbate neuroinflammation and related neuropathologies [189]. These data suggest that an underlying physiological sexual dimorphism exists at the level of the regulation of inflammation and immune functions, however, the biological foundations of this phenomenon are yet to be clarified.

Sexual differences in glial functions affect ion channel regulation, apoptotic signaling cascades and autophagy, activation of microglia, neuron–glia responses to chemical and physical insults (e.g., changes in ionic balance), and mitochondrial toxicity [178,190]. Interestingly, microglia present gender-based differences in cell numbers, brain distribution, and cellular functions. For example, microglia isolated from female mice possess higher expression of IL-4 and IL-10 receptors, and exhibit elevated production of IL-4, after IL-10 cytokine treatment relative to those isolated from male animals [149]. Conversely, microglia from male mice have higher expression of Iba1 at sites of immune activation, and resting glia have increased Neuroglobin and arginase-1 levels in males as compared to female mice [149]. These sex differences have been hypothesized to be definitive in the increased male vulnerability in SCZ. This may be in part because microglia and their inflammatory and “neurosculpting” functions are involved in the developmental process of sexual differentiation [191], and also due to the increased number of activated microglia in the male brain under normal developmental circumstances [192,193]. In support of this idea, a recent study found significant differences in the distribution, arborization, cellular stress profile, and synaptic modulatory capacity of microglia in the hippocampus of male versus female mouse offspring following innate immune activation (inflammatory challenge) [194]. The same group also detected increased expression of inflammatory pathway genes in the cerebral cortex and hippocampus of male mice challenged with the same innate immune trigger [194]. Overall, these results suggest that the increased incidence of SCZ in males might involve increased and dysregulated microglial responses to prenatal immune insults. Aberrant microglial functions may also involve abnormal regulation of synaptic pruning, neuron–glia crosstalk, neuroinflammation, oxidative stress, and autophagy, which may contribute to the spectrum of cognitive impairments in SCZ in both sexes. Available literature data on biological sex-related differences in neuroinflammation in BIP are scarce. However, in the light of the presented reports regarding sexual dimorphism in inflammatory glial functions in SCZ, results warrant further investigation into other psychotic disorders as well.

## 6. Neurotransmission in Schizophrenia and Bipolar Disorder

Neurotransmitters are the chemicals responsible for controlling brain functions; they exert this action by carrying, boosting, and balancing signals between neurons. Dysfunctions in neurotransmission systems—including neurotransmitters, their receptors, transporters, and all intracellular processes coupled to the activation of receptors for neurotransmitters and growth factors—lead to deficits in neuronal transmission at chemical synapses with severe consequences on brain processes and cognitive functions. Mounting evidence [27,93,195,196,197,198] indicates that an imbalance of major excitatory and inhibitory neurotransmitter systems in the brain (dopamine, serotonin, glutamate and GABA) underlies the cognitive deficits and symptoms observed in both SCZ and BIP patients.

This section summarizes the main neurochemical hypotheses of SCZ and BIP and highlights the most recent studies in support of these theories. In addition, it reviews the role of astrocytes in neurotransmission and their functional deficits in psychotic disorders. Finally, it provides an overview of the most recent findings on the critical role of sex hormones in balancing the excitatory and inhibitory transmission pathways with rescue properties on brain functions.

### 6.1. Altered Neurotransmission Pathways in SCZ and BIP

#### 6.1.1. The Monoamine Theory

The monoamine hypothesis of SCZ and mood disorders postulates that an imbalance in monoaminergic neurotransmission in the CNS is causally related to the clinical features of SCZ and BIP [199,200,201]. Evidence suggests that dopamine levels are elevated in certain areas of the brain, resulting in over-stimulation and excess sensory information that correlates with difficulties in concentration, thought process, reality orientation, feelings, and behavior [202,203]. In BIP, pharmacological evidence indicates that, mainly due to the altered availability of dopamine D2/3 receptors and transporters [204], intrinsic dysregulation in the homoeostatic modulation of dopaminergic function would increase dopaminergic transmission in mania and the converse would happen in depression [205,206,207]. In SCZ, the dopaminergic hypothesis is supported by the effectiveness of all known antipsychotic drugs in alleviating the positive symptoms of SCZ by blocking striatal dopamine D2 receptors [202,203,208]. More recent studies based on genetic animal models with dysregulated dopaminergic neurotransmission [209] and pharmacological approaches in patients have demonstrated hyperexcitability and deficits in sensory gating that overlap with the behavioral deficits of SCZ and BIP patients [210].

Recent evidence suggests that abnormalities in serotonin activity also play an important role in psychiatric disorders [198,211]. Postmortem studies indicate altered binding density of serotonin (5-HT) receptor subtypes 5-HT_1A_ and 5-HT_2A_ and abnormal levels of 5-hydroxyindoleacetic acid (5HIAA) and its precursor tryptophan [212,213,214]. Abnormal expression of several noradrenaline (NE)- and 5-HT-related genes have been also found in BIP patients [215]. These alterations have been implicated in the disruption of glutamate signaling which leads to decreased action potential generation, hypometabolism, synaptic atrophy, and grey matter loss [212].

#### 6.1.2. The Glutamatergic and the GABAergic Hypotheses

More recent models hypothesize that a dysregulation of glutamatergic and GABAergic neurotransmission have a primary role in the pathophysiology of SCZ [27,210] and BIP [216,217], where dopaminergic imbalance would be a secondary effect [218].

In the CNS, glutamate is the principal excitatory neurotransmitter and mediates the fast-excitatory transmission by activation of the ionotropic glutamate receptors alpha-amino-3-hydroxy-5-methyl-4-isoxazole propionic acid (AMPA), kainate, and N-methyl-D-aspartate receptors (NMDARs), as well as metabotropic glutamate receptors (mGluRs). The dominant glutamatergic hypothesis of SCZ postulates that psychotic symptoms and cognitive impairments are due to a hypofunction of NMDARs, which leads to excessive glutamate release and hyper-glutamatergic functions. In support of this theory, NMDAR antagonists, like phencyclidine (PCP) and ketamine, produce negative symptoms and cognitive dysfunction closely resembling SCZ phenotypes in healthy subjects [219,220,221]. In animal models, suppression of NMDAR function by pharmacological or genetic approaches led to SCZ-like behaviors [222]. NMDAR hypofunction reduces the function of GABAergic interneurons [223], which then leads to increased pyramidal cell firing due to disinhibition [224] and increased downstream glutamatergic activity resulting in excitotoxicity and cell death [225]. Glutamate dysregulation also causes dysregulation of cortical dopamine [225,226] and contributes to disrupted GABAergic neurotransmission in the brain [227,228].

GABA is the predominant inhibitory neurotransmitter in the CNS. It is synthesized from glutamate by the enzyme glutamate decarboxylase (GAD). Studies using postmortembrain tissue from SCZ patients and animal models have indicated a deficiency of GABA synthesis resulting from reduced transcription of *GAD67* within PV-immunoreactive cortical neurons, as well as a reduction in the subpopulation of GABAergic interneurons positive for PV [227,229,230,231,232,233]. Additionally, several studies have indicated that glutamate-mediated disruption of fast-spiking PV GABAergic interneuron pathways play a major role in the generation of synchronicity and gamma oscillation in the brain. Subjects with SCZ exhibit altered gamma-band activity that often correlates with symptoms and cognitive deficit, suggesting that GABAergic transmission and PV interneurons are responsible for the cognitive decline in SCZ [27,234,235].

Considerable evidence also implicates imbalanced glutamate and GABA neurotransmission in the biochemical pathophysiology of BIP. In support of this, lithium, the first-line treatment for mood stabilization in patients with BIP, reduces neuronal excitation by reducing dopamine levels, downregulating NMDAR-mediated release of glutamate, increasing GABAergic neurotransmission and attenuating calcium metabolism and signaling in the brain [236,237,238,239].

### 6.2. The Role of Astrocytes in Neurotransmission

Astrocytes regulate neurotransmission by altering the concentration of transmitters both inside and outside the synaptic gap. Astrocytes are the primary locus for the biosynthesis of glutamate from glucose, and regulate glutamate metabolism, transport, uptake, and transmission [240]. In psychiatric disorders, the activity of glutamine synthetase (GS) is reduced; this builds up intracellular glutamate in astrocytes, leading to reduced uptake capacities of excitatory amino acid transporters (EAATs). It also reduces the production of glutamine by astrocytic GS which is one of the major sources for maintaining synaptic vesicle content of GABA in inhibitory interneurons. Decreases in GS expression levels, loss of interneurons and deficient cortical GABA synthesis result in increased network hyperexcitability and produce spontaneous recurrent seizures [241]. Deficits in EAAT functions cause persistently increased glutamate levels in the brains of SCZ patients and can increase the susceptibility of the brain to injury and cell death. Finally, astrocytes influence NMDARs via D-serine, an agonist of the glycine-binding site of NMDA receptors, and kynurenic acid (KYNA), an endogenous antagonist [148]. Therefore, an impairment in the synthesis and accumulation of D-serine may lead to disrupted NMDAR activity and cognitive deficits in SCZ [240].

Activation of astrocytes in SCZ patients [148] can trigger the excessive release of pro-inflammatory agents (cytokines, interleukins, and chemokines) which damages neurons, causes alteration of oligodendrocytes [242], and inhibits GABAergic interneurons [243]. Morphological changes of astrocytes can alter neuronal networks contributing to the development of SCZ symptoms. Finally, in SCZ, glia lose the ability to form essential components of the extracellular matrix with serious consequences for the stabilization and maturation of synapses and neuronal connections, and the balance of neurotransmitter systems [148].

### 6.3. The Effect of Estrogen on Neurotransmission

Mounting evidence indicates that E2 exerts profound effects on brain functions by acting on the pathways of several neurotransmitters [26,48]. In particular, E2 has been shown to be a potent neuromodulator, having positive effects on cognitive processes, including learning and memory, as well as mood [244]. The beneficial effects of E2 in SCZ and BIP were proposed to occur through the modulation of monoamine transmitter systems such as dopamine and serotonin [245,246,247].

E2 has been reported to have potent serotonin-modulating properties including the regulation of the level of neurotransmitter synthesis, the degradation of 5-HT and the density and binding of 5-HT receptors [248]. Recent findings suggest that E2 stimulates a significant increase in the density of 5-HT_2A_ binding sites in areas of the brain important for the control of mood, mental state, cognition, emotion and behavior [245,249]. In support of this, E2 therapy and 5-HT uptake blockers such as fluoxetine have proven good efficacy in treating the depressive symptoms of the premenstrual syndrome. Interestingly, E2 treatment has been observed to decrease mRNA levels related to serotonergic neurotransmission. For instance, E2 and progesterone treatment were shown to alter the expression of several genes within the rat dorsal raphe nucleus that are involved in serotonergic transmission, including the postsynaptic 5-HT_2A_ receptor, the presynaptic SERT and vesicular monoamine transporter (VMAT2) [246]. Additionally, the 5-HT_1B_ autoreceptor mRNA in the dorsal raphe [247] and the MAO-A mRNA and activity [250] are decreased after estrogen treatment.

Sex hormones can also impact dopaminergic neurotransmission via a multitude of mechanisms (synthesis, release, turnover and degradation, pre-and postsynaptic receptors, transporters) that still require further investigation. Despite the conflicting findings, most experts agree that E2 has an overall facilitating effect on dopaminergic neurotransmission [248]. In support of this theory, data indicate that E2 induces a substantial increase in dopamine D2 receptors in the striatum [249]. In a different study, the protective action of E2 in SCZ was mediated by downregulating the D2 receptor sensitivity, producing an effect similar to that of antipsychotic drugs blocking DA neurotransmission. Both reduce the probability of the occurrence of schizophrenic symptoms or the triggering of a psychotic episode by enhancing the vulnerability threshold [246].

More recent findings indicate that the beneficial effects of E2 on cognitive functions in SCZ and BIP patients also result from its effects on the glutamatergic and GABAergic transmission systems [250]. E2 probably exerts its beneficial effects via the modulation of glutamatergic synapses and neuronal excitability [244]. The impact of ovarian hormones on the glutamatergic system has been studied extensively and both stimulatory and inhibitory effects have been reported [248,251]. Whereas progesterone mainly impacts non-NMDA receptors [251], the mechanisms underlying the effects of E2 on cognition are related to NMDA glutamate receptors. E2 treatment promotes an increase in NMDA receptor subunit expression, binding sites and neuronal sensitivity to synaptic input mediated by NMDA glutamate receptors in the hippocampus [252,253]. Synaptic functioning and neuronal excitability in the hippocampus are also subject to long-term and short-term excitatory modulation by E2 [254]. An example comes from a work where E2 increased spine density in the hippocampus and the number of excitatory synapses, where long-term potentiation (LTP) can occur, leading to enhanced excitatory postsynaptic potentials (EPSPs) and LTP [255,256,257]. E2 also facilitates the spine maturation process. A study has shown that, in a cellular model of synapse loss relevant for SCZ and BIP, E2 can restore the number of excitatory synapses [250]. Several other animal and in vitro studies have shown that E2 may induce an increase in dendritic spine density through different mechanisms, including the suppression of GABAergic inhibitory neurotransmission [258], as well as the upregulation of AMPA [259,260] and NMDA receptors [252] in the hippocampus and prefrontal cortex (PFC) [261].

With respect to the effect of hormones on the glutamatergic signaling pathway, E2 may protect against NMDA-mediated excitotoxicity [251], apoptosis [252] and oxidative effects [254,255]. In addition to these cellular effects, the interaction between E2 and glutamate can affect cognitive domains such as working memory and executive function under harmful conditions [248].

Sex hormones also interact with the GABAergic neurotransmission pathway. In particular, E2 seems to suppress GABA inhibitory inputs [258]. Recent findings indicate that E2 may alter the functioning of GABAergic and PV-expressing neurons by binding to ER-α or ER-β [262], which then act as transcription factors to modulate the expression of PV [263] and the synthesis of GABA, or to increase the number of postsynaptic GABAergic receptors or their binding affinity. This would compensate for deficits of GAD67 and GABA synthesis that represent typical phenotypes of SCZ and BIP [228]. E2 can also modulate the dynamics of surface GABA receptors. Acute exposure of neurons to E2 leads to destabilization of the specialized populations of GABA_A_ receptors and the inhibitory scaffold protein gephyrin at inhibitory synapses, leading to reductions in the efficacy of GABAergic inhibition via a postsynaptic mechanism. Because this regulatory mechanism largely affects the fast-synaptic inhibition in the adult brain, it may also have profound effects on the excitatory-inhibitory balance which in turn influences synaptic plasticity and cognitive functions. It is important to emphasize that virtually all the promising neuromodulatory effects of E2 described here have resulted from in vitro and in vivo studies using animal models or cell lineages, which may not fully recapitulate the complexity of human brain circuitry. Human disease models based on iPSCs are cutting edge technology that makes it possible to revisit these important hypotheses on E2 neuromodulation in a more advanced disease model (see more details on Section 8).

## 7. Transcriptional Effects of Estrogen

Given that the primary effect of ER activation is gene expression regulation, many studies have investigated the complex and intricate transcriptional changes that are induced by E2 binding to its receptors. The first global and hypothesis-free studies of E2 effects on gene expression were published almost two decades ago and relied on serial analysis of gene expression (SAGE) and microarray technologies [264]. Although most of these global experiments focused on the impact of E2 exposure in breast cancer cell lines, they nevertheless revealed extensive regulatory effects of the hormone encompassing a multitude of molecular pathways and mechanisms [265,266,267,268,269]. Interestingly, these studies suggested that E2 downregulates many target genes that are either known to inhibit the cell cycle or are pro-apoptotic, which is in agreement with the view that E2 promotes cell survival and proliferation [264].

More recently, transcriptome-wide studies have also been carried out in CNS tissues and cell types. One of the earliest of these studies investigated the transcriptional impact of E2 in the primate prefrontal cortex and found that E2 significantly regulated 40 genes by two-fold or more, including the neuronal activation marker c-FOS, which was increased by 2.3-fold and localized specifically to pyramidal neurons [270]. Another study, which examined the global effects of E2 and ER agonists in the rat frontal cortex, showed that E2 had a significant impact on the expression of 16 genes, with an over-representation of genes involved in neuroinflammatory processes [271]. These findings also indicated that glial cells are potential targets of E2 action as several of the downregulated genes, such as macrophage expressed-1 (*MPEG1*), CX3C chemokine receptor 1 (*CX3CR1*), cluster of differentiation molecule 11b (*CD11B*), and toll-like receptor-4 (*TLR4*), are predominantly expressed in microglia [271]. This suggests that E2 may suppress microglia reactivity and thereby protect against inflammation-mediated neurotoxicity [271,272].

Both clinical and preclinical studies provide strong evidence that E2 has positive effects on cognition [27,273], which is relevant to the cognitive dysfunction observed in BIP and SCZ, since memory impairment is a core feature of these disorders [274]. This has led many researchers to investigate the extent to which E2 treatment modulates gene expression in the mammalian hippocampus. Aenlle et al. (2009) treated ovariectomized mice with E2 for five weeks and performed microarray analysis on the hippocampal tissue [275]. They found that 187 genes exhibited altered expression in response to E2. This set of genes was enriched for functions that are critical for the growth and protection of hippocampal cells, including transcription, cell signaling pathways, cell growth, and lipid and protein metabolism [275]. A similar study looked at global expression in the mouse hippocampus after acute E2 exposure and identified 73 genes that were upregulated and 53 that were downregulated, of which 17 and 6 genes are known to be involved in learning and memory, respectively [276]. Several of these were validated with qPCR and confirmed at the protein level, including the known E2 responsive heat shock protein gene HSP70 and the synaptosome-specific gene *SNAP25*, which is required for neurotransmitter release [276].

Widespread transcriptional effects of E2 have also been reported in cultured CNS cells. Cultured neurons and glial cells obtained from the developing human brain and exposed to E2 for seven days were found to alter the expression of 199 genes [277]. These genes comprise several functional categories that are of interest with respect to the neuroprotective and neuromodulatory effects of E2 in the brain, such as cell differentiation, cell cycle regulation, signal transduction, apoptosis, and ion channels and transporters [277], indicating that both neurons and glial cells are important mediators of estrogen action.

In addition to the abovementioned global experiments, several studies have focused on specific genes and pathways that have been associated with E2 through other lines of evidence. These targeted studies have generally confirmed the associations of interest at the transcriptional level, showing for example that E2 increases the mRNA expression levels of *BDNF* [278,279], as well as several dopamine receptor subtypes in mammalian brain tissues [280]. Moreover, E2 was found to upregulate the expression of catalytic subunits of the mitochondrial respiratory chain complexes I, III, IV, and V in primary astroglia, the major energy supplier in the CNS, providing further evidence that increased ATP production and, in consequence, reduced ROS levels in astrocytes could represent a mechanism by which E2 protects neurons from cell death under neurotoxic conditions [120].

Taken together, these global and targeted gene expression studies demonstrate that E2 exerts complex transcriptional effects in the mammalian brain involving multiple cellular processes and pathways. These processes could act as molecular mediators of the neuroprotective effects attributed to E2 and its potential role in the development of CNS disorders like SCZ and BIP. Although no study has yet investigated the specific effects of E2 on neuron–glia interactions at the gene expression level, these findings do indicate that both cell types are targeted by E2 and that it may protect neurons both directly and through glial cell-mediated mechanisms. Table 1 displays a list of effects of E2 on mitochondrial metabolism, inflammation, and neurotransmission which contribute to the neuroprotective action of the hormone.

## 8. Induced Pluripotent Stem Cell (iPSC) Models to Study Neuron–Glia Interactions

The discovery of iPSCs [288] has offered new and promising prospects for studying and treating complex diseases. Some of the main applications of iPSCs include the fields of regenerative medicine, disease modeling, drug screening, toxicity assessments and clinical cell therapies. Like any other cellular model, iPSCs also hold some limitations, like genetic instability, tumorigenic threat after transplant, and lack of proper epigenetic resetting after iPSC differentiation [289,290,291]. Nevertheless, iPSC-derived models still embody one of the best platforms to study polygenic disorders such as SCZ and BIP, since animal models cannot fully reproduce the complexity of polygenic disease phenotypes. In fact, during the past years iPSCs have been demonstrated to be a useful tool as in vitro models to study neurodegenerative disorders such as amyotrophic lateral sclerosis [292,293], Huntington’s disease [294] or Alzheimer’s disease [295,296], and neurodevelopmental disorders like SCZ [30,98,297] and BIP [31,298], among other diseases. IPSC technology can be useful in the identification of specific genes involved in drug-response in highly heritable disorders, such as SCZ and BIP, by comparing transcriptomic data of iPSC-neurons from patients that are responsive to a specific drug treatment with data from those that are non-responsive. Nakazawa and colleagues (2017) used this strategy by producing TUJ1^+^ neurons differentiated from iPSCs derived from a rare case of homozygotic twins with SCZ in which one was responsive, and the other non-responsive to clozapine treatment. The iPSC-neurons were treated with clozapine which resulted in different expression levels of cell adhesion molecule genes, such as *CDH8*, *DS3* and protocadherin genes, between the responsive and non-responsive twin. Alterations in these molecules in the brain can potentially result in dendritic and synaptic changes, resulting in alterations in the neural circuitry and clinical symptoms [299]. Interestingly, one study presented preliminary data about the beneficial role of E2 in iPSC-neurons, where they showed that treatment with 10 nM E2 for 24 h results in an increase in dendritic branching [286]. However, the iPSCs were derived from only one healthy male individual, and no additional investigation was done with iPSC-derived neurons from patients. In addition, it has been shown that the systemic administration of estradiol-2-benzoate (E2B) into a rat model of Parkinson’s disease facilitated the integration of grafted dopaminergic neurons derived from human iPSC into the host neuronal circuit [287].

A further development within iPSC research was the establishment of cellular co-culture models, allowing cell–cell interactions accurately mimicking the in vivo environment of astrocyte–neuron coordination [297,300,301,302,303,304]. However, one of the biggest issues in human neuron–astrocyte co-culture research has been the usage of rodent astrocytes together with human iPSC derived neurons, which might be counterproductive due to known differences between human and rodent glial gene expression [305,306,307]. Furthermore, trying to recreate in vitro the astroglia–neuron interactions present in mental disorders like SCZ and BIP with rodent astrocytes does not appear to be an ideal experimental setup, given the differences between human and rodent brain cells. Astrocyte complexity greatly increased during hominid evolution and species differences should be taken into account when studying mechanisms underlying human neurological disorders [308,309,310]. In fact, it has been proposed that human cortical evolution was accompanied by an increasing complexity in the function and morphology of astrocytes, with an expansion of their roles in synaptic modulation and cortical circuitry [309]. For example, in a recently published study using a humanized glial chimeric mouse model, iPSC-derived glial progenitor cells (GPC) from childhood-onset SCZ patients were implanted into myelin-deficient mice. Interestingly, this resulted in compromised glial maturation with reduced white matter expansion of the GPC, astrocytic differentiation delay and abnormal cellular morphologies [311]. Another work has established a human cortical iPSC system for investigating astrocyte-to-neuron interactions. Cortical astrocytes and neurons were generated from a common pool of OTX2^+^ cortical radial glia progenitors. The iPSC-astrocytes expressed well-known astrocyte-specific genes, astrocyte-specific membrane properties, gap-junction coupling, and showed capacity to respond to neurotransmitters such as glutamate and ATP. In such co-cultures the cortical iPSC-derived astrocytes accelerated the maturation of cortical iPSC-derived neurons by increasing electrical excitability and synaptic network activity. This was reinforced by the fact that the astrocytes expressed genes encoding extracellular enzymes that promote synapse structure and maturation, glutamatergic transmission, cell adhesion, neurite outgrowth, and synaptic development [312]. Thus, this model provides a source of human astrocytes and neurons suitable for astrocyte–neuron signaling studies in neurodevelopmental pathologies like SCZ and BIP.

Recently, the development of iPSC-derived 3D systems have raised great promise in the field of in vitro brain disease modeling [32,313,314,315,316,317,318], where different strategies are being used to generate brain organoids representing different regions analogous to the midbrain [319], hindbrain [320,321] and forebrain [314,317,318], adding an extra layer of complexity in comparison with 2D iPSC-derived models. However, limitations of brain organoids include the lack of vascularization and a high variability between replicates [32,314]. In addition, some of the neuronal population in 3D systems are not well specified and contain a mixture of mature and progenitor cells, without a precise control over cell maturity [314,317,318,322,323]. Nevertheless, some attempts have been made to solve the lack of vascularization by transplanting the brain organoids into mice brains [324], and to reduce variability using novel protocols [317,318]. Therefore, despite these limitations, brain organoids still embody the best current models for the study of neuron–glia interactions, since they recapitulate brain architecture and function. After long periods of differentiation and maturation, they are able to innately develop astrocytes and microglia [317,318,325] and recreate complex oscillatory waves similar to the ones found in the neonatal brain EEG signals [326]. Here we highlight the potential benefits of studying the effects of E2 in neuron–glia interactions using in vitro models of BIP and SCZ based on iPSCs derived from patients, whilst the advantages of using cerebral organoids to study the influence of steroid hormones in brain development have been reviewed elsewhere [327]. The different cellular models based on iPSCs and their applicability for genetic, metabolic, and electrophysiological studies are shown in the Figure 1.

## 9. Conclusions and Perspectives

SCZ and BIP are severe mental disorders which are difficult to treat, causing life-long morbidity in a large number of young individuals worldwide. SCZ and BIP share not only clinical symptoms, but also genetic factors and pathophysiological mechanisms affecting energy metabolism, immune status, neurotransmission, and neuron–glia interactions. As these systems are interrelated and involved in crosstalk with one another, it is difficult to identify alterations that are intrinsic, and appear first in the course of the disease, from the ones that are secondary and appear later, as a result of primary alterations. Sexual dimorphism seems to be present in both disorders, with women being, in general, less affected than men, and evidence suggests that the female hormone E2 could be an important factor responsible for this difference. In fact, a number of studies have shown neuroprotective effects of E2 upon different brain cell insults in several animal and cellular models of disorders involving sexual dimorphism. Beneficial effects of E2 have been attributed to its influence on energy metabolism, immune system, and neurotransmission, all affected in SCZ and BIP. Indeed, E2 and SERMs have been successfully used as adjuvant drugs in BIP and SCZ clinical trials, highlighting the importance of understanding the precise mechanisms underlying beneficial effects of E2 in BIP and SCZ in order to identify novel and sex-specific drug targets for SCZ and BIP. Indeed, meta-analyses of randomized controlled trials have shown that the SERM raloxifen is effective in improving total symptom severity in SCZ spectrum disorders [328], as well as the Positive and Negative Syndrome Scale (PANSS) total psychopathology, positive and negative symptoms, and general psychopathology scores in postmenopausal women with SCZ [329]. Clinical trials have also confirmed that the SERM tamoxifen has been effective to treat episodes of mania in BIP patients [330]. Growing evidence has shown that besides autonomous effects of E2 on neurons, the non-autonomous effects of E2 via glial cells, such as astrocytes and microglia, are essential for E2 neuromodulatory effects. Disease models of BIP and SCZ have gradually moved from rodents, human peripheral blood and postmortembrain tissue into patient-derived iPSCs differentiated into brain cells and organoids, which offer several advantages and avoid many of the limitations associated with other model systems. In particular, when studying the mechanisms underlying E2 beneficial properties, one should be cautious of extrapolating results from animals to humans due to interspecies differences in the expression of the different ER types across different brain regions. In addition, since the neuromodulatory role of E2 involve neuron–glia interactions, iPSC-derived cells and 3D cultures are better suited for this purpose. Different types of iPSC-derived neural cells can grow as monocultures, mixed co-cultures, brain organoids, or specific brain region spheroids and be treated with E2 before being tested for a number of parameters, including gene expression, as well as parameters of energy metabolism, inflammation, and electrophysiology. IPSC-based models are a relatively new technology that will continue to evolve in terms of cellular composition and network, allowing the measurement of an increasing number of parameters. In this context, testing the mechanisms underlying beneficial or cell-damaging effects of sex hormones in SCZ and BIP, using an iPSC-based models, arises as a promising new research field. Additionally, unveiling, in detail, the mechanisms by which E2 displays beneficial effects to the CNS will lead to the identification of potential targets for the development and discovery of new drugs to treat several neurological disorders.

## Figures and Tables

**Figure 1 cells-10-00209-f001:**
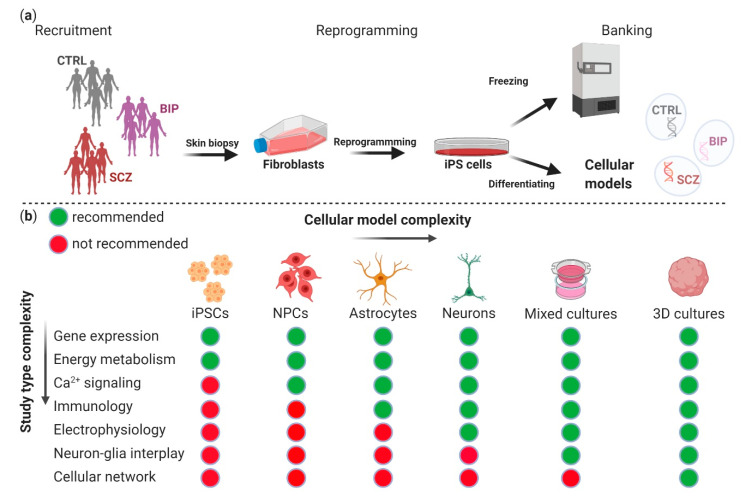
The potential of iPSCmodels to study schizophrenia (SCZ) and bipolar disorder (BIP). (**a**) Scheme showing the process of recruitment, reprogramming of skin fibroblasts into iPSCs, cell banking, and generation of cellular models for healthy controls and patients with SCZ and BIP. (**b**) IPSC-based models listed by complexity and their suitability for different types of studies relevant for SCZ and BIP. CTRL, healthy controls; NPCs, neural progenitor cells.

**Table 1 cells-10-00209-t001:** List of studies addressing the effects of estrogen in different models.

Experimental Model	Preparation	E2 Effects	Receptor Involved	Reference
Ovariectomized mice	Whole brain mitochondria	↑ activity of complex IV (COX); ↑ expression of nuDNA subunit COX IV, mtDNA subunit COX I, and of the antioxidant enzymes MnSOD, PrdxV, and glutathione peroxidase; ↓ lipoperoxidation; ↑ maximal mitochondrial capacity	ERα or Erβ	Irwin et al., 2012 [115]
Human lens cells	Cell culture	↓ oxidative stress; ↓ mitochondrial membrane permeability	ERβ1	Flynn et al., 2008 [116]
Human pituitary tumor cells	Cell culture	↑ expression of the COX II subunit	N.I.	Van Itallie et al., 1988 [117]
Rats	Hippocampal tissue	↑ expression of the COX III subunit; ↑ activity of COX	N.I.	Bettini et al., 1992 [118]
Several human cell lines	Cell culture	↑ expression of the COX VII subunit	N.I.	Watanabe et al., 1998 [119]
Mouse cortical and mesencephalic astrocytes	Cell culture	↑ expression of catalytic subunits of the ETC complexes I, III, IV and V; ↑ activity of ATP synthase; ↑ mtDNA/nuDNA ratio	N.I.	Araújo et al., 2008 [120]
MCF-7 and H1793 cells	Cell culture	↑ expression of NRF-1; ↑ mitochondrial biogenesis	N.I.	Mattingly et al., 2008 [121]
Mouse astrocytes	Cell culture	↑ transcription of mitochondrial fusion/fission proteins	N.I.	Arnold et al., 2008 [123]
HeLa cells	Cell culture	↑ MCU-mediated mitochondrial Ca^2+^ uptake	N.I.	Lobatón et al., 2005 [124]
Ischemia-perfusion in Female rats	Brain tissue	↑ expression of anti-apoptotic proteins (e.g., Bcl-2)	Erα	Zhang et al., 2017 [125]
SH-SY5Y cells	Cell culture	↑ expression of the antioxidant proteins thioredoxin and MnSOD	N.I.	Chiueh et al., 2003 [126]
HT22 cells, mouse hippocampal and neocortical, and C6 cells	Cell culture	↑ expression of glutathione	N.I.	Schmidt et al., 2002 [127]
Rat cortical neurons submitted to glutamatergic excitotoxicity	Cell culture	↓ apoptosis; prevention of cytochrome c release; ↓ expression of caspase-3	N.I.	Zhang et al., 2005 [128]
Pubertal period female mice	Ventromedial hypothalamus	↑ microglial response to LPS	N.I.	Velez-Perez et al., 2020 [185]
Rat cortical neurons transfected with mutated DISC1	Cell culture	↑ spine density and synaptic proteins; ↓ DISC1 aggregates	N.I.	Erli et al., 2020 [244]
Rat cortical neurons submitted to glutamatergic excitotoxicity	Cell culture	↓ release of lactate dehydrogenase	Classical estrogen receptors	Singer et al., 1996 [281]
Ovariectomized rats	Hypothalamic arcuate nucleus	↑ number of Bcl-2-immunoreactive neurons	N.I.	Garcia-Segura et al., 1998 [282]
Rat hippocampal neurons submitted to oxidative stress inducers	Cell and organotypic culture	Prevention of peroxide accumulation	Independent on any ER activation	Behl et al., 1997 [283]
Ovariectomized rats	Several brain regions	Prevention of serotonin-receptor loss due to ovariectomy	N.I.	Cyr et al., 2000 [284]
Ovariectomized rats	CA1-region hippocampal neurons	↑ excitability, independent of NMDA receptors	N.I.	Wong et al., 1992 [254]
Male adult rats	Hippocampal slices	↑ synaptic responses by a mechanism dependent on integrin activation and signaling	Erβ	Wang et al., 2016 [257]
Rats and mice	Cell culture and brain slices	↓ amplitude of inhibitory synaptic currents; destabilization of GABAARs and gephyrin at inhibitory synapses	N.I.	Mukherjee et al., 2017 [285]
Rats	Frontal cortex organotypic slice cultures	↑ cortical expression of parvalbumin in both deep and superficial layers	N.I.	Ross and Porter, 2002 [263]
Ovariectomized monkeys	Dorsolateral prefrontal cortex	Regulated 40 genes, including ↑ expression of C-FOS and ↓ expression of E2F1 and TFIIB mRNA and protein	N.I.	Wang et al., 2004 [270]
Ovariectomized rats	Frontal cortex	Regulated 16 genes including ↓ expression of complement C3 and C4b, Ccl2, Tgfb1, macrophage expressed gene *Mpeg1*, RT1-Aw2, Cx3cr1, Fcgr2b, Cd11b, Tlr4 and Tlr9, defensin Np4, RatNP-3b, IgG-2a, Il6 and the ER gene *Esr1*	ERα and Erβ	Sárvári et al., 2011 [271]
Ovariectomized mice	Hippocampus	Regulated 187 genes, mostly of them involved in transcription, cell signaling, cell growth, and lipid and protein metabolism	N.I.	Aenlle et al., 2009 [275]
Female mice	Dorsal hippocampus	Alteration in the expression of 204 genes, of which 23 are involved with learning/memory; changes in the content of the proteins Hsp70, Igfbp2, Actn4, Tubb2a, and Snap25	N.I.	Pechenino and Frick, 2009 [276]
Human fetus	Neuron and glial cell cultures	Altered expression of 199 genes, many implicated in cell differentiation, cell cycle regulation, signal transduction, apoptosis, and ion channels and transporters	N.I.	Csöregh et al., 2009 [277]
Gonadectomized male rats	Hippocampus	↑ levels of BDNF mRNA and protein	Erα	Solum and Handa, 2002 [278]
Intact and ovariectomized female rats	Hippocampus, cortex and spinal cord	↑ expression of the *BDNF* gene	N.I.	Allen and McCarson, 2005 [279]
Ovariectomized rats	Amygdala, hypothalamus, nucleus accumbens, midbrain, and ventral tegmental area	↑ levels of dopamine and serotonin receptors mRNA; ↓ levels of ERα and ERβ mRNA	N.I.	Zhou et al., 2002 [280]
Human cells	iPSC-derived forebrain neurons	↑ number of dendritic branches	N.I.	Shum et al., 2015 [286]
Human cells in a rat model of Parkinson’s disease	iPSC-derived dopaminergic neurons	Activation of integrin α5β1 in the rat striatum; ↑ integration of grafted neurons into host striatum	N.I.	Nishimura et al., 2016 [287]

*↑*, increased; *↓*, decreased; ETC, electron transport chain; BDNF, brain-derived neurotrophic factor; iPSC, induced pluripotent stem cell; LPS, lipopolysaccharide; MCU, mitochondrial calcium uniporter; N.I., not informed in the study; NMDA, N-methyl-D-aspartate.

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
