# Peer review of "Using iPSC Models to Understand the Role of Estrogen in Neuron–Glia Interactions in Schizophrenia and Bipolar Disorder"

_cells, 2021, doi:10.3390/cells10020209_

Round 1

Reviewer 1 Report

This is a timely and comprehensive review that provides mechanistic insights into the role of estrogen in neuron-glia interactions in schizophrenia and bipolar disorder. Before, publishing the paper, it is recommended that the authors try to address the following points:

  1. The reader might be curious whether any GWASs provided evidence for the association between estrogen signalling and schizophrenia or bipolar disorder.
  2. The authors should discuss current evidence regarding the efficacy of treatments that target estrogen signalling in schizophrenia and bipolar disorder. The following meta-analyses might be helpful for quantitative synthesis in this field: de Boer et al. (NPJ Schizophr 2018,4(1):1) and Zhu et al. (Schizophr Res 2018, 197:288-293).

Author Response

Response to reviewers` comments and corresponding alterations in manuscript:

Reviewers` comments are written in normal type

Authors’ responses are written in red

Corresponding alterations in the manuscript are written in italics

Reviewer #1 Comments for the Author

This is a timely and comprehensive review that provides mechanistic insights into the role of estrogen in neuron-glia interactions in schizophrenia and bipolar disorder. Before publishing the paper, it is recommended that the authors try to address the following points:

  1. The reader might be curious whether any GWASs provided evidence for the association between estrogen signalling and schizophrenia or bipolar disorder.

Response to comment:

We appreciate this comment. Indeed, we added an extra paragraph (lines 199-206) at the end of section 3 (Genetics of schizophrenia and bipolar disorder) mentioning GWAS associations between estrogen signalling and schizophrenia and bipolar disorder.

Corresponding alterations in the manuscript:

Section 3 (lines 199-206):

GWAS from European descents [74] and from the Korean population [47] have identified associations of SNPs related to E2 biosynthesis and ERs with SCZ. More recent GWAS have identified that a genetic risk (E2 polygenic risk score) for higher plasma E2 is negatively associated with hippocampal volume, but not with an increased risk of major depressive disorder or post-partum depression [75]. However, the largest GWAS of anxiety traits to date was able to identify genome-wide significant associations near genes involved with the ERα (ESR1) [76]. Besides, the SNP rs2144025 has been associated with increased ESR1 mRNA levels in prefrontal cortex from subjects with BIP and SCZ and appears to modulate traits in behavioral disorders [77].”

  1. The authors should discuss current evidence regarding the efficacy of treatments that target estrogen signalling in schizophrenia and bipolar disorder. The following meta-analyses might be helpful for quantitative synthesis in this field: de Boer et al. (NPJ Schizophr 2018, 4(1)) and Zhu et al. (Schizophr Res 2018, 197:288-293).

Response to comment:

We appreciate this comment. We followed the Reviewer’s suggestion and added the references suggested by the Reviewer, de Boer et al. (NPJ Schizophr 2018 ,4(1)) and Zhu et al. (Schizophr Res 2018, 197:288-293) to section 9  (Conclusions and Perspectives). We also added to the same section the reference Palacios et al. (J Psychopharmacol. 2019 Feb;33(2):177-184), about the use of the SERM tamoxifen to treat patients with bipolar disorder.

Corresponding alterations in the manuscript:

Section 9 (lines 857-863):

Indeed, meta-analyses of randomized controlled trials have shown that the SERM raloxifen is effective in improving total symptom severity in SCZ spectrum disorders [332] as well as the Positive and Negative Syndrome Scale (PANSS) total psychopathology, positive and negative symptoms and general psychopathology scores in postmenopausal women with SCZ [333]. Clinical trials have also confirmed that the SERM tamoxifen has been effective to treat episodes of mania in BIP patients [334].

Please find the references in the attached pdf file

Reviewer 2 Report

I suggest that the authors should adopt a more selective and critical approach to the problem, according to which one develops an increasingly sharp elimination of successive attempts to get at the core. There are of course tomes written on this question, but an eminently readable account is Popper by Bryan Magee of the Fontana Modern Masters series. or Miles Weatherall's Scientific Method' 1968.

Author Response

Response to reviewers` comments and corresponding alterations in the manuscript:

Reviewers` comments are written in normal type

Authors’ responses are written in red

Corresponding alterations in the manuscript are written in italics

Reviewer #2 Comments for the Author

I suggest that the authors should adopt a more selective and critical approach to the problem, according to which one develops an increasingly sharp elimination of successive attempts to get at the core. There are of course tomes written on this question, but an eminently readable account is Popper by Bryan Magee of the Fontana Modern Masters series or Miles Weatherall's Scientific Method' 1968.

Response to comment:

We thank the Reviewer for this insightful comment. We agree that a well-established principle in scientific inquiry is to use a focused, inductive method to achieve the best explanatory power within the theoretical frame of enquiry. This principle applies primarily when formulating hypotheses or theoretical advancements that build on pre-existing and proven theories. These must include a rigorous, step-by-step process of successive verifications pertaining to the given hypothesis (as in e.g. Kuhn, 1962) and formulating further model elements to explore the domain of the observed phenomenon (e.g. Lakatos, 1980, Philosophical Papers, Vol. 1). Popper has, in fact, a very different approach to this method by denying the existence of any verification procedures (e.g. Popper K., 1959, The Logic of Scientific Discovery, see chapters 1-4). Instead of the classical approach, he put the emphasis on the principle of falsification, i.e. a negative test would inevitably necessitate the rejection of the given hypothesis/theory. We agree with the Reviewer that Popper’s method can be relevant in some contexts to clear up the explanatory frame during hypothesis making, and e.g. to differentiate scientifically relevant hypotheses from non-relevant (pseudoscientific) ones. However, the suggested sources (Magee, 1985; Weatherall, 1968) mostly discuss the possibility of the application of falsification.

We have carefully evaluated the findings in the field and selected the results based on a heuristic approach and knowledge in the field. Further, we did not intend to perform theoretical verification of the available scientific hypotheses in the field by applying iPSC-based model systems (reviewed in our paper). The primary goal of our review is to build a rationale for selecting a set of existing hypotheses in psychiatry that can now be tested in experimental settings, by applying novel stem cell technology. We suggest how preferably more advanced approaches reflecting the inherent methodological potential of these iPSC/organoid systems can be used in novel studies targeting major unanswered questions in psychiatry and neuroendocrinology. However, we do agree with the Reviewer that it is possible to sharpen the focus and build more logical arguments for the proposed research plans. We have now revised the text accordingly.

We re-wrote and sharpened the abstract of the manuscript in order to objectively reach the most important messages of the review.

Corresponding alterations in the manuscript:

Abstract (lines 17-34):

“Schizophrenia (SCZ) and bipolar disorder (BIP) are severe mental disorders with a considerable disease burden worldwide due to early age of onset, chronicity, and lack of efficient treatments or prevention strategies. Whilst our current knowledge is that SCZ and BIP are highly heritable and share common pathophysiological mechanisms associated with cellular signalling, neurotransmission, energy metabolism and neuroinflammation, the development of novel therapies has been hampered by the unavailability of appropriate models to identify novel targetable pathomechanisms. Recent data suggest that neuron-glia interactions are disturbed in SCZ and BIP, and are modulated by estrogen (E2). However, most of the knowledge we have so far on the neuromodulatory effects of E2 came from studies on animal models and human cell lines, and may not accurately reflect many processes occurring exclusively in the human brain. Thus, here we highlight the advantages of using induced pluripotent stem cell (iPSC) models to revisit studies of mechanisms underlying beneficial effects of E2 in human brain cells. A better understanding of these mechanisms opens the opportunity to identify putative targets of novel therapeutic agents for SCZ and BIP. In this review, we first summarize the literature on the molecular mechanisms involved in SCZ and BIP pathology and the beneficial effects of E2 on neuron-glia interactions. Then, we briefly present the most recent developments in the iPSC field, emphasizing the potential of using patient-derived iPSCs as more relevant models to study the effects of E2 on neuron-glia interactions.”

We added a paragraph (lines 376-382) at the end of section 4.3 (Influence of estrogen on the mitochondrial metabolism) emphasizing the rationale of testing effects of estrogen on mitochondrial function in human brain cells.

Corresponding alterations in the manuscript:

Section 4.3 (lines 376-382):

“Taken together, the aforementioned works suggest a number of ways by which E2 may improve mitochondrial activity in several regions of the CNS. However, studies of modulatory mechanisms of E2 on mitochondria of human brain cells are still lacking, but hold the potential to confirm many of the routes identified in cell lines and rodent models. Such studies may also identify new mechanisms underlying the beneficial effects of E2 in several neurological conditions where gender differences exist, such as mental disorders.”

We also added a paragraph at the end of section 6.3 (The effect of estrogen on neurotransmission) calling the attention to the fact that most of the evidence about the effects of estrogen on neurotransmission comes from studies of animal models and cell lines, which do not fully recapitulate the complexity of the human brain. We indicate that in section 8 more details about iPSC models are provided.

Section 6.3 (lines 688-693):

“It is important to emphasize that virtually all the promising neuromodulatory effects of E2 described here have resulted from in vitro and in vivo studies using animal models or cell lineages, which may not fully recapitulate the complexity of human brain circuitry. Human disease models based on iPSCs are cutting edge technology that makes it possible to revisit these important hypotheses on E2 neuromodulation in a more advanced disease model (see more details on section 8).”

Reviewer 3 Report

Careful and extensive review of the latest work on the role of estrogen in neuron-glia interactions in schizophrenia and bipolar disorder, including the iPSC model. I have no major objections to the text.

Author Response

Response to reviewers` comments and corresponding alterations in manuscript:

Reviewers` comments are written in normal type

Authors’ responses are written in red

Corresponding alterations in the manuscript are written in italics

Reviewer #3 Comments for the Author

Careful and extensive review of the latest work on the role of estrogen in neuron-glia interactions in schizophrenia and bipolar disorder, including the iPSC model. I have no major objections to the text.

Response to comment:

We thank the Reviewer for recognizing our efforts.
